# Cladribine Reduces Trans-Endothelial Migration of Memory T Cells across an In Vitro Blood–Brain Barrier

**DOI:** 10.3390/jcm11206006

**Published:** 2022-10-12

**Authors:** Rachel K. Ford, Pierre Juillard, Simon Hawke, Georges E. Grau, Felix Marsh-Wakefield

**Affiliations:** 1Vascular Immunology Unit, School of Medical Sciences, Faculty of Medicine and Health, The University of Sydney, Sydney, NSW 2006, Australia; 2Central West Neurology and Neurosurgery, Orange, NSW 2800, Australia; 3Liver Injury and Cancer Program, Centenary Institute, Sydney, NSW 2050, Australia; 4Human Cancer and Viral Immunology Laboratory, The University of Sydney, Sydney, NSW 2006, Australia

**Keywords:** blood–brain barrier, cladribine, multiple sclerosis, T cells, spectral flow cytometry, trans-endothelial migration

## Abstract

Multiple sclerosis (MS) is a chronic, demyelinating disease of the central nervous system (CNS) induced by immune dysregulation. Cladribine has been championed for its clinical efficacy with relatively minor side effects in treating MS. Although it is proposed that cladribine exerts an anti-migratory effect on lymphocytes at the blood–brain barrier (BBB) in addition to its lymphocyte-depleting and modulating effects, this has not been properly studied. Here, we aimed to determine if cladribine treatment influences trans-endothelial migration of T cell subsets across an inflamed BBB. Human brain endothelial cells stimulated with pro-inflammatory cytokines were used to mimic the BBB. Peripheral blood mononuclear cells were obtained from healthy controls, untreated and cladribine-treated MS patients. The trans-endothelial migration of CD4^+^ effector memory T (T^EM^) and CD8^+^ central memory T (T^CM^) cells was reduced in cladribine-treated MS patients. CD28 expression was decreased on both CD4^+^ T^EM^ and CD8^+^ T^CM^ cells, suggesting lowered peripheral activation of these cells thereby maintaining the integrity of the BBB. In addition, these cells have likely reconstituted following cladribine treatment, revealing a long-term anti-migratory effect. These results highlight new mechanisms by which cladribine acts to control MS pathogenesis.

## 1. Introduction

Cladribine is known to be a selective immune-reconstitution therapy used for the treatment of multiple sclerosis (MS) [1]. It is a deoxyadenosine analogue pro-drug which exerts its primary mechanism via the preferential depletion of lymphocytes [2]. Cladribine has been championed for its safety profile and ‘pulse-like’ course delivered in 2 short cycles, generating lasting immune reconstitution and potentially drug-free remission in relapsing-remitting MS [3,4]. The observed lasting clinical benefits associated with cladribine, even after cessation of drug exposure has led to the hypothesis that cladribine exerts other immunomodulatory effects beyond lymphocyte depletion [5]. Previous studies have demonstrated cladribine’s ability to alter cytokine profiles to an anti-inflammatory phenotype and inhibit T cell activation [6,7]. However, the proposed effect of cladribine on immune cell migration across the blood–brain barrier (BBB), despite representing a crucial event in MS pathogenesis, has not been studied extensively.

Trans-endothelial migration of immune cells across the BBB into the brain parenchyma is one of the earliest and most important events in MS pathogenesis [8]. Consequently, blockade of leukocyte migration across the BBB is shown to be therapeutically effective for reducing MS disease activity [9]. Kopadze and colleagues studied cladribine’s effect on cell migration in vitro, finding a reduced migration of cladribine-stimulated CD4^+^ and CD8^+^ T cells isolated from MS patients ex vivo [10]. However, more work is needed to determine the effect on more defined T cell subsets. 

The present study investigates the migration profiles of T cell subsets across the BBB in vitro to further explore cladribine’s proposed anti-migratory effects. CD4^+^ effector memory T (T^EM^) and CD8^+^ central memory T cells (T^CM^) showed reduced trans-endothelial migration across the BBB in cladribine-treated MS patients. Analysis of key cell migration and activation markers revealed increased CD38 expression and decreased CD28 expression on migrated CD4^+^ T^EM^ cells 24 months post-cladribine which may result in reduced BBB dysfunction whereby fewer cells can permeate across. Finally, the peripheral blood concentrations of CD4^+^ T^EM^ and CD8^+^ T^CM^ cell subsets were shown to decrease 4 months post-cladribine. We hypothesise that following cladribine-mediated depletion of lymphocytes, reconstituted CD4^+^ T^EM^ and CD8^+^ T^CM^ cell subsets have an impaired ability to migrate across the BBB. In this context, the decreased functional capacity of these cells is, in part, a consequence of modified surface activation marker expression. These results provide a deeper understanding of lymphocyte migration during neuroinflammation and unveil new therapeutic effects for cladribine. 

## 2. Materials and Methods

### 2.1. Study Participants 

Patients with relapsing-remitting type MS as diagnosed using the McDonald criteria (McDonald et al., 2010) were recruited from Central West Neurology and Neurosurgery in Orange, NSW and the Brain and Mind Centre at the University of Sydney in Sydney, NSW. Ethical consent for the study was received from the Research Integrity and Ethics Administration of the University of Sydney (humans ethics code 2018/377). 

Blood samples were collected from untreated MS patients, cladribine-treated patients and age-/sex-matched healthy volunteers (Table 1). Untreated MS patients had no treatment and active disease as defined by clinical and MRI activity within the last 3 months. Blood from oral cladribine (Mavenclad©, Merck)-treated patients was sampled prior to and 4 and 24 months following treatment; total initial dose 1.75 mg/kg administered in two 5-day courses one month apart and repeated 12 months later (total cumulative dose over two years 3.5 mg/kg). 

### 2.2. PBMC Isolation from Whole Blood

Fresh whole blood was sampled from patients and healthy controls using EDTA collection tubes and kept at room temperature (15–20 °C). Blood was diluted with equal amounts of Phosphate-Buffered Saline without calcium and magnesium (PBS, Sigma Chemical Co., St. Louis, MO, USA). Peripheral blood mononuclear cells (PBMC) were obtained through Ficoll density gradient separation using 15 mL Ficoll-PAQUE (GE Healthcare Pharmacia, GE Healthcare, Chicago, IL, USA) overlaid with 25–32 mL diluted blood and centrifuged at 400× *g* for 30 min at 20 °C with brakes off. 

PBMC were collected from the plasma Ficoll-PAQUE interface and diluted with PBS-2%FBS followed by 2 centrifugations at 120× *g* for 10 min at room temperature. Cells were resuspended in PBS-2%FBS and counted as the fresh cell sample using a Counter II Automatic Cell Counter (Thermo Fisher Scientific, North Ryde, NSW, Australia). The viability was calculated using Trypan blue staining before use in the transmigration assay. Fresh PBMC underwent flow cytometry staining or was used in the transmigration assay (described below). 

### 2.3. Transmigration Assay

Transmigration assays were performed using a modified Boyden^®^ chamber protocol, as previously described by Hawke and colleagues [11]. In brief, a 2-compartment migration system (Appendix A) was used with an upper and lower chamber separated by a polycarbonate membrane with 3 μm sized pores (Costar Pharma, Smithfield, NSW, Australia), coated in collagen to mimic the extracellular matrix with a confluent layer of human cerebral microvascular endothelial cells (hCMEC/D3 referred to as HBEC) to mimic the endothelial cell barrier growing on it. 

HBEC (Institute Cochin, Paris, France) were seeded onto collagen-coated inserts (105 cells/cm^2^) and after 20 min, 2.6 mL of warmed 1% antibiotic antimycotic solution (AA, Sigma Chemical Co., St. Louis, MO, USA) in complete EBM-2 medium (EBM-2 medium supplemented with 5% FBS, 5 μg/mL ascorbic acid (Sigma, St. Louis, MO, USA), 1.4 μmol/L hydrocortisone (Sigma), CDLC (1:100 dilution; Life technologies, Carlsbad, CA, USA), 10 mmol/L HEPES (Sigma) and 1 ng/mL β-FGF (Sigma)) was added to the lower compartment of the cell culture inserts and then incubated at 37 °C and 5% CO_2_ overnight. 

The medium in the lower chambers and within the inserts was refreshed on days 2 and 3 by transferring each insert into a new 6-well plate filled with 2.6 mL of warmed 1% AA in complete EBM-2 medium and then adding 1.5 mL of the same solution into the insert before incubation overnight at 37 °C and 5% CO_2_.

On day 4, in order to stimulate inflammation of the HBEC monolayer, each insert was transferred into a new 6-well plate filled with 2.6 mL of warmed 1% AA in complete EBM-2 medium supplemented with TNF (5 ng/mL, Peprotech, Rocky Hill, NJ, USA) and IFN-γ (10 ng/mL, Peprotech) and then 1.5 mL of the same solution was added into the insert before incubation overnight at 37 °C and 5% CO_2_. Cytokines were washed off prior to co-culture with PBMCs.

On day 5, the integrity of the HBEC monolayer was assessed at least 3 h prior to performing the transmigration assay by using Wheat Germ Agglutinin Alexa Fluor™ 555 conjugate (WGA, Invitrogen, Thermo Fisher Scientific, Waltham, MA, USA). WGA binds selectively to N-acetylglucosamine and N-acetylneuraminic (sialic) acid residues and permitted staining of the whole HBEC monolayer. Each insert was transferred into a new 6-well plate filled with 2.6 mL of warmed WGA solution (5 µg/mL) and then 1 mL of the same solution was added into the insert before incubating inserts at room temperature in the dark for 10 min. Inserts were then washed twice with PBS, followed by adding 1.5 mL of warmed 1% AA in complete EBM-2 medium into each insert. Using a fluorescent microscope, the HBEC monolayer was checked for the absence of holes and multiple layers, ensuring a complete monolayer was formed. Inserts were then washed twice with PBS before each insert was transferred into a new 6-well plate filled with 2.6 mL of warmed 1% AA in complete EBM-2 medium and then 1.5 mL of the same solution was added into each insert. 

After confirmation of membrane integrity and quality, each insert was then transferred into a new 6-well plate filled with 2.6 mL of warmed 3% FBS in RPMI 1640 medium (referred to as RPMI-3% FBS, Sigma, St. Louis, MO, USA) and incubated for at least 3 h at 37 °C and 5% CO_2_. Then, 1.5 mL of isolated PBMC (1.56 × 10^6^/mL) resuspended in RPMI-3% FBS was added to the cell culture inserts and incubated at 37 °C and 5% CO_2_ for 14–18 h. 

On day 6, the non-migrated cells in the upper compartment of the wells were carefully collected without touching the monolayer. The cell culture insert was removed and migrated cells in the lower well compartment were collected. Non-migrated and migrated cells and were counted using a Counter II Automatic Cell Counter (Thermo Fisher Scientific, North Ryde, NSW, Australia) and percentage viability was calculated using Trypan blue staining before analysis by flow cytometry. 

### 2.4. Spectral Flow Cytometry

On the same day as the isolation of PBMC (fresh) or collection from the migration assay (non-migrated and migrated cells), flow cytometric analysis was performed. 

PBMC were centrifuged for 5 min at 500× *g* at room temperature. Each pellet was resuspended in FACS buffer (PBS supplemented with 0.5% BSA (Sigma Chemical Co., St. Louis, MO, USA), and 2 mM EDTA (AMRESCO Inc, Solon, OH, USA)), incubated for 20 min at 4 °C in a staining mix consisting of FACS buffer with fluorescently conjugated antibodies (Table 2), washed twice and then fixed in 4% paraformaldehyde (PFA) for 20 min at room temperature. Flow cytometry was performed using an Aurora 5-laser flow cytometer (Cytek, Fremont, CA, USA). All flow cytometric analysis was performed using FlowJo software v10.7 (BD, Macquarie Park, NSW, Australia). 

### 2.5. Gating Strategy 

Samples were first gated based on their forward-scatter-height (FSC-H) and time to obtain a consistent representative sample. Total PBMC were then gated based on their forward-scatter-area (FSC-A) and side-scatter-area (SSC-A) followed by gating single cells by excluding doublets/triplets using FSC-A and FSC-H cytograms (Appendix A). Notably, CD8^+^ T cells were divided, according to their level of CD8 expression, into high CD8 expressing (CD8^high^) and low CD8 expressing (CD8^low^) cells. 

### 2.6. Statistical Analysis

Statistical analyses were performed using Graph Pad Prism 9 software (GraphPad, San Diego, CA, USA). 

To evaluate the migratory capacity of T cell subsets, the number of migrated and non-migrated cells was quantified. The ratio of migrated over non-migrated cells was calculated. For visualisation, the logarithm base 2 of this ratio was calculated for each subset. A positive value indicates active migration, whilst a negative value indicates no migration. The value represents a base 2 fold change, where a value of 1 indicates twice as many cells in the migrated chamber. A value of −1 (negative 1) means twice as many cells were in the non-migrated chamber.

A similar method was utilised for comparing the median fluorescence intensity (MFI) for each marker between migrated and non-migrated T cell subsets. The MFI for each marker was calculated, and the ratio of the migrated compared to non-migrated was calculated. The logarithm base 2 of this ratio was calculated for each subset. A positive value indicates higher expression of the marker in the migrated population, whilst a negative value indicates a higher expression in the non-migrated population.

To compare more than two groups, a Kruskal–Wallis nonparametric one-way ANOVA with a Dunn’s multiple comparisons test was performed. For comparisons within a group, after a logarithm of 2-fold change of the ratio of paired migrated to non-migrated lymphocyte populations, a one sample nonparametric Wilcoxon signed-rank test with Pratt method was used, comparing the median of test groups to a hypothetical value of 0. A *p*-value ≤ 0.1 was shown on all plots.

## 3. Results

### 3.1. CD4^+^ T^EM^ and CD8^+^ T^CM^ Cells Have Inhibited Transmigration across the BBB in Cladribine-Treated MS Patients

In all trans-endothelial migration experiments, the level of cells that did not migrate (cells remaining in the upper chamber), and those that migrated (cells found in the bottom chamber), were phenotyped and measured as total cell numbers. The logarithm of 2-fold change was calculated using the number of migrated cells compared to non-migrated cells (Log_2_(M/NM)). A positive value indicates more cells migrated whilst a negative value indicates fewer cells migrated. 

In contrast to healthy control (HC) and untreated MS groups, CD4^+^ T^EM^ cells at 4 months post-cladribine (Clad 4M) did not exhibit a preference for migration across the stimulated HBEC monolayer due to fewer CD4^+^ T^EM^ cells found in the lower chamber relative to the top chamber (Figure 1(Ai)). Across all groups, naïve CD4^+^ T cells (T^Naive^) did not exhibit a preference for migration (Figure 1(Aii)). 

Amongst HC, CD8^high^ T^CM^ cells showed an overall preference for migration. In contrast, CD8^high^ T^CM^ cells from untreated MS patients had a reduced preference for migration. Although there was no noticeable trend in the migration for CD8^high^ T^CM^ cells at Clad 4M, there was a further reduced tendency for migration at 24 months post-cladribine (Clad 24M) (Figure 1(Bi)). Like their counterpart, CD8^low^ T^CM^ cells showed a reduced tendency to migrate in the untreated MS patients and a further reduced tendency at Clad 24M (Figure 1(Ci)). Across all groups, CD8^high^ and CD8^low^ T^EM^ cells showed an overall preference for migration across the stimulated HBEC monolayer (Figure 1(B–Cii)). 

The migration of other T cell subsets can be found in Appendix A.

### 3.2. CD38 Expression Increased on Migrated CD4^+^ T^EM^ Cells Whilst CD28 Expression Decreased on Migrated CD4^+^ T^EM^ and CD8^+^ T^CM^ Cells in Cladribine-Treated MS Patients

After determining the inhibited migration of CD4^+^ T^EM^ and CD8^+^ T^CM^ cells across the BBB in vitro, the expression of migration and activation markers CD49d, CD38, and CD28 were analysed for both the migrated cells in the lower well and non-migrated cells in the upper well.

CD49d expression was higher on migrated CD4^+^ T^EM^ and CD8^high^ and CD8^low^ T^CM^ cells compared to non-migrated cells (Figure 2(Ai–Aiii)). For migrated CD4^+^ T^EM^ cells, CD38 was significantly increased in the Clad 24M group relative to Clad 4M and HC (Figure 2(Bi)). For CD8^+^ T^CM^ cells, there were no noticeable changes in CD38 expression (Figure 2(Bii,Biii)). Conversely, there was a significant reduction in CD28 expression on migrated CD4^+^ T^EM^ cells in the Clad 24M compared to untreated MS patients and HC (Figure 2(Ci)). CD28 expression was significantly reduced on CD8^low^ T^CM^ cells in Clad 24M compared to HC (Figure 2(Ciii)).

### 3.3. Cladribine Depletes Circulating CD4^+^ T^EM^ and CD8^+^ T^CM^ Cells

The peripheral blood concentration of CD4^+^ T^EM^ cells was significantly reduced in Clad 4M compared to untreated MS patients (Figure 3A). CD8^high^ T^CM^ cells were significantly reduced in Clad 4M compared to HC (Figure 3B). Furthermore, CD8^low^ T^CM^ cells were significantly reduced in Clad 4M compared to both HC and untreated MS patients (Figure 3C). The peripheral blood concentration of other T cell subsets can be found in Appendix A.

## 4. Discussion

This study had two primary objectives: firstly, to identify any differences in the migratory behaviour of T cell subsets across the BBB between healthy controls, untreated MS patients and cladribine-treated MS patients; and secondly, to analyse the expression of surface markers on migrated cells which may contribute to their migratory behaviour. Using an in vitro inflamed BBB, CD4^+^ T^EM^ and CD8^+^ T^CM^ cells from cladribine-treated MS patients had an impaired transmigratory ability with altered CD28 and CD38 expression. These findings may relate to the consequences of immune reconstitution following cladribine-mediated lymphocyte depletion and may explain cladribine’s long-term effects.

In untreated MS patients, most IFN-γ is produced by CD4^+^ T^EM^ cells [12], which holds true in healthy individuals [13,14]. IFN-γ can upregulate the expression of adhesion molecules such as VCAM-1 and ICAM-1 and was shown to favour the trans-endothelial migration of CD4^+^ T^EM^ cells [15]. Moreover, activated myelin oligodendrocyte glycoprotein (MOG)-specific T cells pre-treated with cladribine produce significantly less IFN-γ [16]. The decrease in IFN-γ production following cladribine may result in reduced CD4^+^ T^EM^ cell trans-endothelial migration.

There was also a decrease in the trans-endothelial migration of CD8^high^ and CD8^low^ T^CM^ cells in cladribine-treated MS patients 24 months after initial dose. CD8^low^ T cells are distinct from CD8^high^ T cells in flow cytometric analysis of peripheral blood T cells and have exhibited different functional capabilities including an enhanced ability to execute cytotoxicity and altered cytokine production [17]. In MS, CD8^low^ T cells are decreased in frequency in peripheral blood and appear as a target of immunomodulatory therapy [18]. Notwithstanding these phenotypic and functional differences, both CD8^high^ and CD8^low^ T^CM^ cells exhibited similar patterns of trans-endothelial migration in cladribine-treated MS patients. CD8^+^ T cells have shown an enhanced ability to migrate across the stimulated BBB in vitro upon exposure to interleukin-15 (IL-15) [19]. In the presence of IL-15, activated CD8^+^ T cells are also more likely to acquire a T^CM^ phenotype [20]. Furthermore, MS patients have increased IL-15 levels which is hypothesised to be a consequence of a higher proportion of B cells and monocytes expressing IL-15 relative to non-MS patients [19]. Therefore, the decreased migratory ability of CD8^+^ T^CM^ cells may be a consequence of cladribine’s depletion of IL-15 levels either directly or indirectly through modulation of the B cell population. It would be of great interest to determine cladribine’s effect on IL-15 which has yet to be investigated [6,7].

Migrated memory T cells expressed higher CD49d compared to their non-migrated counterparts, with no change in expression following cladribine. CD49d is a subunit of VLA-4 on lymphocytes and facilitates trans-endothelial migration across the BBB [21]. Moser and colleagues found no differences in CD49d expression on circulating CD4^+^ or CD8^+^ T cells from cladribine-treated MS patients over 24 months [22]. These results suggest that cladribine exerts its anti-migratory effects independently of CD49d despite the molecule’s central role in trans-endothelial migration. This notion is further supported by the observation that natalizumab, an anti-CD49d antibody treatment for MS, and not cladribine, is associated with increased risk of progressive multifocal leukoencephalopathy (PML), resulting from wide-spread inhibition of lymphocyte migration across the BBB [23].

CD38 expression was increased on migrated CD4^+^ T^EM^ cells relative to non-migrated cells 24 months after cladribine compared to both healthy controls and 4 months post-cladribine. CD38 was first identified as an activation marker and later reported to be an adhesion molecule, interacting with endothelial CD31 [24,25]. Whilst a role for CD38 in supporting the trafficking of immune cells during infection-induced inflammation has been established [26,27], the association of CD38 with the process of T cell trans-endothelial migration across the BBB is yet to be elucidated. Animal models suggest that CD38 promotes disease progression in MS [28,29]. CD38 deficiency in mice reduced experimental autoimmune encephalitis (EAE) severity and suppressed glial activation, axonal damage, and demyelination which was shown to be a result of lowered MOG-specific T cell responses indicating T cell priming and proliferation effects [28,29,30]. Cladribine’s inhibition of CD4^+^ T^EM^ cell migration and the fact that migrated cells express higher levels of CD38 suggests cladribine’s selective depletion of CD38-expressing CD4^+^ T^EM^ cells. In addition, CD38 may play a role in trans-endothelial migration across the BBB via its interaction with CD31, as shown previously whereby CD38 was found to be essential for T cell infiltration in a mouse model of cerebral ischemia [31]. Previous studies on cladribine’s lymphocyte depletion effect demonstrated that median CD4^+^ T cell counts recovered to threshold values by week 96 post initial dose [32]. This recovered T cell population may consist of a lower number of CD38-expressing CD4^+^ T^EM^. Further investigation into the CD38 expression on circulating CD4^+^ T^EM^ cells is therefore warranted.

In contrast to CD38 expression, CD28 expression was decreased on CD4^+^ T^EM^ and CD8^low^ T^CM^ cells in cladribine-treated MS patients. CD28 is the major costimulatory receptor and plays a central role in the second signal of the “two-signal model” of T cell activation following antigen presentation [33]. The role of CD28 in MS pathogenesis has also been established with the blockade of CD28 shown to suppress the onset of EAE in animal models [34,35]. Given that immune activation outside of the CNS precedes dysfunction of the BBB and subsequent immune cell infiltration in MS, it is plausible that lowered CD28 expression on CD4^+^ T^EM^ and CD8^low^ T^CM^ cells after cladribine treatment may contribute to the reduced migratory capacity of these cell subsets [36].

Memory T cells were depleted following cladribine treatment. Cladribine preferentially depletes lymphocytes due to their high DCK:5′NT ratio and mediates its therapeutic effect in MS patients by accumulating within lymphocytes to disrupt cellular metabolism and DNA production, causing apoptosis [37]. In human plasma, the half-life of cladribine varies from 5.7 to 19.7 h [38]. As blood samples were obtained from cladribine-treated MS patients 4 months after initial treatment, there was a 3-month interval between patients’ last administered dose such that any changes at this time were independent of cladribine’s cytotoxicity. Moreover, median CD4^+^ T cell counts were shown to recover to threshold values by week 96 post initial dose with median CD8^+^ T cell counts never dropping below the threshold value [32].

The primary limitation in the current study was that experiments performed in vitro are limited in their ability to reflect the in vivo environment. The attempt to model lymphocyte trans-endothelial migration into the CNS is complicated by the complexity of the brain environment, particularly during inflammation. A triple culture in vitro BBB model has been described with brain endothelial cells, pericytes and astrocytes [39,40], and may more fairly reflect the complex in situ environment.

## 5. Conclusions

The present study is one of the first to provide evidence of cladribine’s effect on T lymphocyte trans-endothelial migration using an in vitro model of the BBB comprised of human cerebral microvascular endothelial cells. In cladribine-treated MS patients, CD4^+^ T^EM^ and CD8^+^ T^CM^ cells had a diminished ability to migrate across the BBB, which can partly be attributed to decreased CD28 expression. Future studies are recommended with the aim to determine cladribine’s effect on cytokines produced by these cell subsets. Here, we provided further insight into the drivers of MS pathogenesis and a secondary mechanism of action for cladribine.

## Figures and Tables

**Figure 1 jcm-11-06006-f001:**
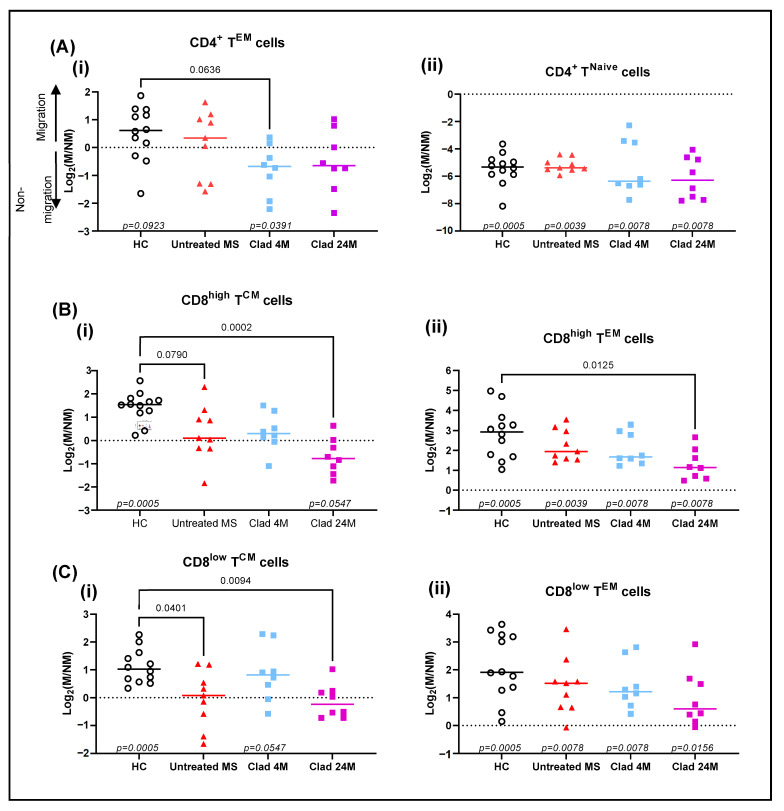
Cladribine inhibits the migration of CD4^+^ T^EM^ cells and CD8^+^ T^CM^ cells across the BBB in vitro. PBMC were isolated from healthy controls (*n* = 12), untreated MS (*n* = 9) and cladribine-treated MS patients at 4 months (*n* = 8) and 24 months (*n* = 8) post initial dose. PBMC were then added to an in vitro model of the stimulated BBB and left to migrate overnight. The logarithm of 2-fold change of migrated to non-migrated cells was calculated. (**A**) The migratory capacity of (**i**) CD4^+^ T^EM^ cells and (**ii**) CD4^+^ T^Naïve^ cells. (**B**) The migratory capacity of (**i**) CD8^high^ T^CM^ cells and (**ii**) The CD8^high^ T^EM^ cells. (**C**) The migratory capacity of (**i**) CD8^low^ T^CM^ cells and (**ii**) CD8^low^ T^EM^ cells. In-group analysis performed using one sample Wilcoxon matched-pairs signed rank test with the Pratt method with *p*-values ≤ 0.1 displayed at the bottom of each graph. Between-group analysis performed using Kruskal–Wallis with Dunn’s multiple comparisons test. Median shown. NM, non-migrated; M, migrated; T^CM^, central memory T cells; T^EM^, effector memory T cells; T^Naive^, naïve T cells; HC healthy controls; MS, multiple sclerosis; BBB, blood–brain barrier; Clad, cladribine; 4M, 4 months post-Clad; 24M, 24 months post-Clad; PBMC, peripheral blood mononuclear cells.

**Figure 2 jcm-11-06006-f002:**
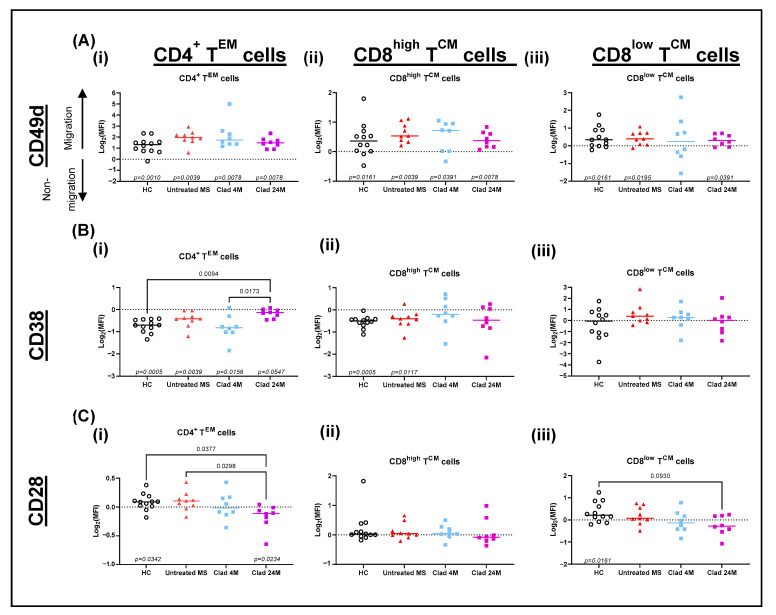
CD28 and CD38 expression are altered on migrated and non-migrated CD4^+^ T^EM^ cells and CD8^+^ T^CM^ cells. The logarithm of 2-fold change of marker expression on migrated to non-migrated cells was calculated. (**A**) CD49d (**B**) CD38 and (**C**) CD28 expression on (**i**) CD4^+^ T^EM^ cells (**ii**) CD8^high^ T^CM^ cells and (**iii**) CD8^low^ T^CM^ cells. In-group analysis performed using a one sample Wilcoxon matched-pairs signed rank test with the Pratt method with *p*-values ≤ 0.1 displayed at the bottom of each graph. Between-group analysis performed using Kruskal–Wallis with Dunn’s multiple comparisons test. Median shown. NM; non-migrated, M; migrated; T^CM^, central memory T cells; T^EM^, effector memory T cells; HC healthy controls; MS, multiple sclerosis; Clad, cladribine; 4M, 4 months post-Clad; 24M, 24 months post-Clad.

**Figure 3 jcm-11-06006-f003:**
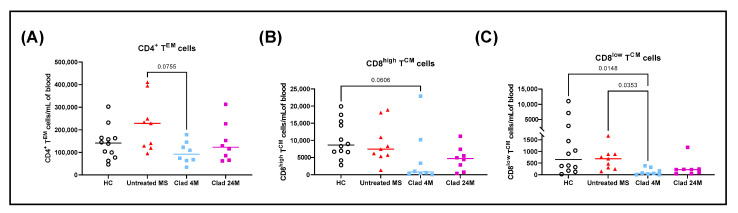
Cladribine depletes CD4^+^ T^EM^ and CD8^+^ T^CM^ cells. Blood was sampled from healthy controls (*n* = 12), untreated MS (*n* = 9) and cladribine-treated MS patients at 4 months (*n* = 8) and 24 months (*n* = 8) post initial dose. Upon sample collection, fresh PBMC were immediately isolated and phenotyped. Each T cell subset was gated on to calculate the proportion of cells multiplied by the cell count and shown as cells per mL of blood. The peripheral blood concentration of (**A**) CD4^+^ T^EM^ cells (**B**) CD8^high^ T^CM^ cells and (**C**) CD8^low^ T^CM^ cells. Kruskal–Wallis with Dunn’s multiple comparisons test. Median and *p*-values ≤ 0.1 are shown. T^CM^, central memory T cells; T^EM^, effector memory T cells; HC healthy controls; MS, multiple sclerosis; Clad, cladribine; 4M, 4 months post-Clad; 24M, 24 months post-Clad; PBMC, peripheral blood mononuclear cells.

**Table 1 jcm-11-06006-t001:** Overview of demographic and clinical data on MS patients and healthy volunteers from which samples were collected for use within this study.

MS Patient ID	Sex (71% Female)	Age * (Median = 41)	Time since MS Diagnosis * (Years)	Time of Blood Sampling since First Cladribine Dose (Months)			Previous Treatment	Months Since Last Treatment Prior to Cladribine
MS01	F	26	3.5	0	4	-	-	-
MS02	F	62	22.0	0	-	-	-	-
MS03	F	53	13.0	0	-	-	-	-
MS04	F	30	0.1	0	-	-	-	-
MS05	F	39	6.4	0	-	-	-	-
MS06	F	69	37.0	0	-	-	-	-
MS07	M	31	0.0	0	-	-	-	-
MS08	M	47	0.0	0	-	-	-	-
MS09	M	37	4.5	0	-	-	-	-
MS10	F	39	10.2	-	4	24	-	-
MS11	F	41	6.2	-	4	24	Fingolimod	2
MS12	F	48	3.5	-	4	-	-	-
MS13	M	35	4.2	-	4	-	-	-
MS14	F	41	1.8	-	4	-	Fingolimod	3
MS15	F	48	21.0	-	4	24	Fingolimod	15
MS16	F	52	18.5	-	4	-	Fingolimod	2
MS17	F	43	8.0	-	-	24	Fingolimod	15
MS18	M	40	4.2	-	-	24	-	-
MS19	F	41	21.0	-	-	24	Oral Prednisone	12
MS20	F	29	4.3	-	-	24	Dimethyl fumarate	2
MS21	M	34	9.8	-	-	24	-	-
**Healthy Control ID**			**Sex (67% Female)**			**Age * (Median = 35)**		
HC01			F			42		
HC02			F			28		
HC03			F			24		
HC04			F			56		
HC05			M			43		
HC06			F			41		
HC07			M			43		
HC08			M			28		
HC09			M			24		
HC10			F			41		
HC11			F			35		
HC12			F			32		

* Age at time of blood sampling.

**Table 2 jcm-11-06006-t002:** List of fluorescently conjugated antibodies and clones used for flow cytometric analysis.

Cell Marker	Fluorochrome	Clone	Company
CD3	Alexa Fluor 532	UCHT1	eBioscience, ThermoFisher Scientific, Waltham, MA, USA
CD4	Alexa Fluor 700	RPA-T4	Biolegend, San Diego, CA, USA
CD8	BV480	RPA-T8	BD Biosciences, Franklin Lakes, NJ, USA
CD16	FITC	3G8	Beckman Coulter, Lane Cove West, NSW, Australia
CD27	BV650	O323	Biolegend
CD28	BV510	CD28.2	Biolegend
CD45RA	PerCP-Cy5.5	HI100	Biolegend
CD49d	PE-Cy5	9F10	Biolegend
CD56	BV750	5.1H11	Biolegend
CD161	PE-Dazzle 594	HP-3G10	Biolegend
CD14	Pacific Blue	M5E2	Biolegend
CD19	BV570	HIB19	Biolegend
CD24	PerCP	ML5	Biolegend
CD38	BV785	HIT2	Biolegend
CD62L	BV711	DREG-56	Biolegend
CD197	PE	G043H7	Beckman Coulter
CD69	BV421	FN50	Biolegend
CD20	Super Bright 436	2H7	eBioscience, ThermoFisher Scientific
CD196	BV605	GO34E3	Biolegend
CD274	PECy7	29E.2A3	Biolegend
GPR56	APC	4C3	Biolegend

## Data Availability

The date presented in this study are available from Merck. Restrictions apply to the availability of these data, which were used under license for this study. Data are available upon request to the authors with the permission of Merck.

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
