# Peer review of "Cladribine Reduces Trans-Endothelial Migration of Memory T Cells across an In Vitro Blood–Brain Barrier"

_jcm, 2022, doi:10.3390/jcm11206006_

Round 1

Reviewer 1 Report

In this manuscript, the authors evaluate whether cladribine reduces T cell migration through the BBB, an effect that along with the already well-documented lymphocyte-depleting and lymphocyte-modulating effects, could further improve the therapeutic outcome in MS.

The authors use PBMCs collected at diverse time points following treatment with cladribine and use an in vitro system to simulate the BBB.  The results presented are scientific sound and support the conclusions reached by the authors. My main criticism has already been raised by the authors: simulation of the BBB has many limits and ideally more than one systems could have been used to collect more data. However, that may be out of the scope of this study. 

Author Response

We thank the reviewer for their kind comments. As stated however, running these experiments on more than one BBB model was outside the scope of this study, essentially for practical reasons: the human brain endothelial cells had to be prepared for each patient’s visit. It would not have been possible, for logistic reasons, to prepare several BBB models in these patient-dependent conditions.

Reviewer 2 Report

In this work, Ford et al analyze the migration of different T cell subsets isolated from MS patients after cladribine treatment. They observed a reduction on the migratory capacity of CD4+ effector memory T cells at 4 months after cladribine administration. On the other hand, they also report the abnormal migration of CD8+ central memory T cells.

The experimental design includes not only the migration assays but also the phenotype of different T cells subsets after the migration. As a result, the authors show a ratio of migrated/non-migrated cells expressed by Log2 in different conditions. However, the understanding of the results using this mathematical formulation is a little bit complicating.

On the whole, it is an interesting paper in the field of MS but there are some points that need to be addressed.

- What is the immunological relevance of CD8hi and CD8low T cell populations? Did you analyse the migration of central memory CD8+ T cells without gating on CD8hi or CD8low? Given the few events of central memory CD8low T cells are observed in supplementary figure 2, I wonder if you could reanalyze the experiments without gating in high and low CD8 expression. Maybe you obtained a more representative population in this way.

- Given that you use the Log2 of the ratio between migrated and non-migrated cells, and it is not easy to understand, you can improve the description of this ratio and the quantification in the material and methods. In the same way, it is not well understood the representation of MFI in the figure 2. Did you calculate the ratio as well? Maybe you can show only the MFI from migrated cell instead of the ratio.

- In the last paragraph of the results you describe the depletion of different T cell populations after cladribine treatment at 4 and 24 months. However, you also reported three different subpopulations. Maybe you can include in this figure the rest of them to give us further information about the effect of cladribine.  

Author Response

What is the immunological relevance of CD8hi and CD8low T cell populations?

More detail on the immunological relevance of CD8high and CD8low T cell populations and their link to multiple sclerosis has been provided in the discussion (lines 297-305). Whilst indeed there are very few central memory T cells which comprise the CD8low T cell population (Supplementary Figure 2), we thought it best to still divide the CD8+ T cell population given the 2 distinct CD8high and CD8low T cell clusters and previously published functional differences.

Given that you use the Log2 of the ratio between migrated and non-migrated cells, and it is not easy to understand, you can improve the description of this ratio and the quantification in the material and methods...

Further clarification has been added in “2.6 Statistical Analysis” of the material and methods (lines 180-192) on the reasoning and meaning behind the Log2 transformation of the ratio between migrated and non-migrated cells to present figures. By presenting the log2 of each ratio in the figure, we were able to more accurately visualize the differences between and within study groups.

In regard to the representation of MFI in figure 2, by calculating the ratio of MFI on migrated cells to non-migrated cells, we were able to obtain the surface marker expression on migrated cells relative to non-migrated cells. In this way, we can provide more information on what types of surface markers may in part contribute to whether or not a cell will migrate across the BBB. The log2 of the ratio was then used in figures to allow for more accurate visualization of differences.

In the last paragraph of the results you describe the depletion of different T cell populations after cladribine treatment at 4 and 24 months. 

  • Supplementary Figure 4 has been added which describes the depletion of other T cell populations after cladribine treatment as 4 and 24 months post initial dose:
    • CD4+ TTEMRA cells (TEMRA, terminally differentiated effector memory cells re-expressing CD45RA)
    • CD4+ TNaive cells
    • CD4+ TCM cells (CM, central memory)
    • CD8high/low TTEMRA cells
    • CD8high/low TNaive cells
    • CD8high/low TEM cells (EM, effector memory)
  • Supplementary Figure 4 has been referenced in the main body of the manuscript (lines 269-270) to supplement Figure 3.

Reviewer 3 Report

The manuscript is well-written and interesting. The Methodology and Discussion is also reasonable and sound. Overall, it provides new piece of evidence that Cladribine impaired lymphocytes migration in addition to the lymphocyte depletion and modulating mechanisms. 

As I am not a statistical expert, I am not sure about the interpretation of the results. Some of the comparisons between groups accept p>0.05 (for example, Fig1Ai, Fig2Ciii, Fig3A and Fig3B) as significant. And this is also presented in the Results. Please consult a statistician that this is correct. Otherwise, the Results and Discussion may be limited to those changes that are statistically significant (p<0.05).   

Author Response

In this paper, a p-value <0.1 was shown on all plots and considered significant. Given the immense human variability encountered when studying the adaptive immune system, we were more lenient with our interpretations of the results on the migratory capacity and surface marker expression on T cell subsets. We were able to discuss key biological differences and not just statistically significant differences in the study.

Reviewer 4 Report

The manuscript “Cladribine reduces trans-endothelial migration of memory T cells across an in vitro blood-brain barrier” studied very interesting topic – the influence of MS therapeutic CLADRIBINE on trans -endothelial migration of T cell subsets across BBB in vitro.

Although the study is entirely in vitro, thus having a very limited “clinically applied” potential, it is indeed well-written and very clearly presented. The Introduction is concise and there is no unnecessary information. Materials and methods are presented in a clear manner, as well as the Results. Similar to Introduction, Discussion is concise and limited to explanation of obtained data and their relevance, as it should. 

Author Response

We thank the reviewer for their kind comments

Round 2

Reviewer 2 Report

Thank you for accepting the recomendations and including the comments.